# PIQPERFECT: DIFFUSION-BASED SAME-IDENTITY FACIAL REPLACEMENT

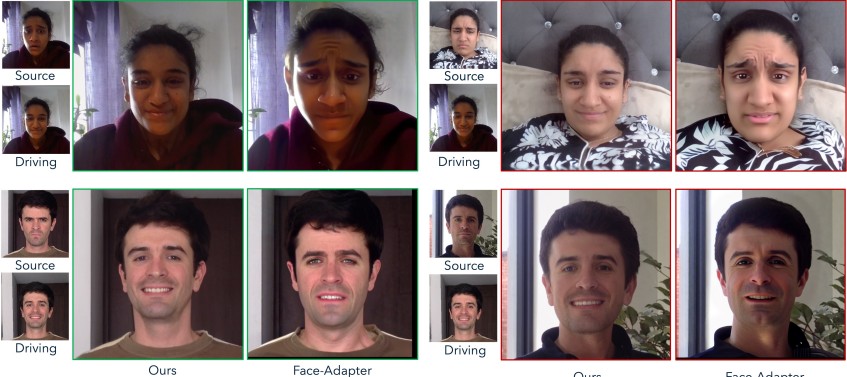

Figure 1: We present PiQPerfect, a diffusion-based identity preserving facial replacement approach. Prior methods like FaceAdapter (Han et al., 2024) tend to alter the facial identity and lose realism on both *in-distribution (ID)* (left) and *out-of-distribution (OOD)* (right) face replacement tasks.

## ABSTRACT

With the growing success of diffusion models in computer vision, we explore their potential for same-identity facial replacement in photographs. Specifically, we propose a diffusion-based method, built on top of a pre-trained text-to-image model, that takes as input a portrait image of a person and a second reference image of the same individual, potentially captured under different conditions. The goal is to seamlessly replace the input face with the reference face, while keeping the background intact. Surprisingly, despite the clear real-world utility of this task, no recently published work has directly addressed face replacement in this specific setting. To support this goal, we construct a large dataset of image pairs depicting the same person under varying facial expressions and poses. Experimental results demonstrate that our approach produces more realistic and identity-consistent results than existing face reenactment models.

## 1 INTRODUCTION

Pasting a person's face onto another image and retouching it using traditional image editing software is a challenging task, particularly when the two images differ in viewpoint, illumination, background, hairstyle, or clothing. In this paper, we propose a diffusion-based approach, built on top of a pre-trained text-to-image model, that can effectively replace a subject's face in a photograph with one that matches the pose and expression from another image of the *same* individual—even in the presence of substantial variation in lighting, pose, or facial expression. A practical application involves replacing an unflattering facial appearance in a group photo with a more favorable one taken from another image, where the person may be smiling or posing more naturally.

An effective solution to this task must preserve the individual's identity, leave the background unaltered, and seamlessly incorporate the reference face into the original image. This is closely related to face reenactment, where a "source" face is morphed to match the pose and expression of a "driving" face. However, most reenactment methods are designed to support *cross-identity* generation,

where the source and driving images may come from different people. While recent reenactment approaches leveraging pre-trained diffusion models, such as Face-Adapter (Han et al., 2024), produce impressive cross-identity results, we observe that these methods often struggle to produce realistic and identity-preserving outputs that faithfully match the driving image. These methods typically rely on the source image alone to retain identity, which becomes a serious limitation in cases with expressions that deviate significantly from the driving image or extreme poses. As shown in Figure 1, our method outperforms such approaches by leveraging the reference image more directly — since both images depict the same person, our model can effectively *composite* the driving face directly into the original photo, rather than attempting to hallucinate it from limited cues.

Our approach shares similarities with early work on automated face swapping (Blanz et al., 2004; Bitouk et al., 2008). Like our method, these classical approaches rely heavily on the reference image by extracting and "pasting" the reference's inner face into the original image after performing re-alignment and relighting. However, these methods typically involve complex pipelines composed of hand-engineered components whereas our approach relies on an end-to-end neural network trained specifically for this task.

Our key contributions are as follows:

- We construct a large-scale dataset of image pairs featuring the same individual under varying facial expressions and poses. We intend to release this dataset publicly.

- We propose a simple yet effective training strategy for extending pretrained text-to-image diffusion model to same-identity face replacement. Compared to prior works, our approach yields realistic, higher-quality outputs with better identity preservation.

- We demonstrate that methods tailored to same-identity generation significantly outperform existing techniques designed for general-purpose, cross-identity face reenactment.

## 2 RELATED WORK

**Same-identity Face Replacement.** Same-identity face replacement has recently garnered increased attention, driven by the growing adoption of generative AI in computer vision applications. Notably, several commercial solutions—including Google's BestTake (Google, 2025), Oppo's AI Perfect Shot (Oppo, 2025), Huawei's AI Best Expression (Huawei, 2025), and OnePlus's AI Best Face (OnePlus, 2025)—incorporate variants of same-identity face replacement. These methods typically utilize a curated set of images *of the same individual* and perform face replacement. Despite its clear real-world utility, to the best of our knowledge, no recently published work has specifically focused on the challenge of producing high-quality, realistic face replacement where the source and driving face are from the same individual. Instead, existing studies have primarily addressed related tasks, such as face reenactment (Hong & Xu, 2023; Tao et al., 2022; Yin et al., 2022; Bounareli et al., 2023) and face swapping (Li et al., 2019; Chen et al., 2020; Zhu et al., 2021; Xu et al., 2022a;b; Liu et al., 2023).

**Face swapping.** Recent face swapping approaches (Li et al., 2019; Chen et al., 2020; Zhu et al., 2021; Xu et al., 2022a;b; Liu et al., 2023) focus on *identity swapping*, where the identity is altered while preserving pose, expression, and background. Since our work focuses on *same-identity* face replacement, these methods are related but not directly applicable to our setup.

**Face Reenactment.** Face reenactment methods are more closely aligned with our task, as they modify a source face to match the pose and expression of a driving image. Most methods are designed for *cross-identity* generations and therefore rely on intermediate representations that discard the identity of the driving image. Existing techniques can be broadly categorized into warping-based and 3DMM-based approaches. Warping-based methods (Siarohin et al., 2019a;b; Hong et al., 2022; Hong & Xu, 2023; Wang et al., 2021; Zhao & Zhang, 2022; Zakharov et al., 2019; Tao et al., 2022) use facial landmarks to transfer motion by warping the source image. These approaches perform well under small motions but degrade significantly with larger pose changes, often producing blurred or distorted outputs due to imprecise motion fields. In contrast, 3DMM-based methods (Kim et al., 2018; Ren et al., 2021; Doukas et al., 2021; Yin et al., 2022) use 3D facial representations (Deng et al., 2019) and are usually more robust to larger pose variations. However, 3DMMs typically lack

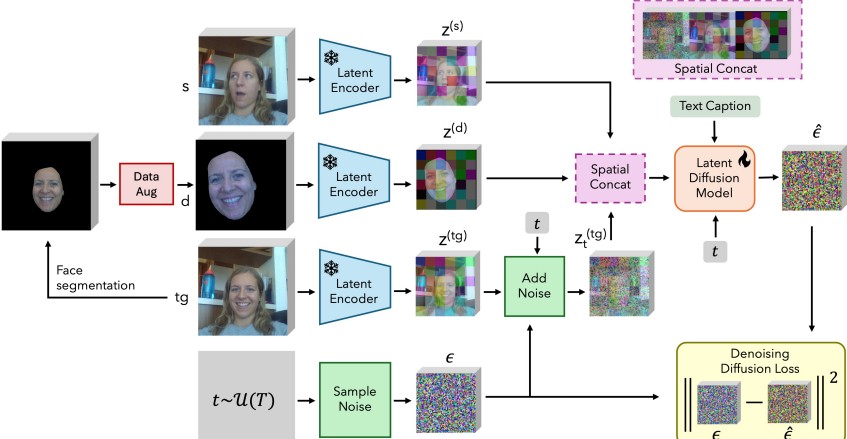

Figure 2: PiQPerfect network overview: Given a source ($s$) and target ($tg$) image pair, we extract the driving image ($d$) by segmenting the face from the target image. To simulate real-world variations, we apply data augmentations such as scaling, rotation, translation, and relighting. The image triplet is then encoded using the SDXL (Podell et al., 2024) VAE to obtain latent representations and concatenated spatially.

high-frequency details, such as hair, teeth, and eye motion. To address this, several works (Yin et al., 2022; Bounareli et al., 2023) integrate adversarial generators like StyleGAN2 (Karras et al., 2020). More recently, diffusion-based face reenactment methods have emerged. FADM (Zeng et al., 2023) combines warping-based facial animation approaches (Siarohin et al., 2019b; Wang et al., 2021) with diffusion-based refinement. Face-Adapter (Han et al., 2024), AniPortrait (Wei et al., 2024b) and DiffusionAct (Bounareli et al., 2024) also leverage pre-trained diffusion models, but rely on keypoints for motion control. A related subcategory includes portrait animation models that require a driving video for motion control. Approaches such as FaceVid2Vid (Wang et al., 2021), X-Portrait (Xie et al., 2024), MegActor (Yang et al., 2024) and LivePortrait (Guo et al., 2024) fall into this class. Like their image-driven counterparts, these methods are designed for cross-identity reenactment and often struggle to preserve identity when the source image contains extreme expressions or poses. In contrast, our approach focuses on *same-identity* face replacement and does not require landmarks, 3DMMs, or auxiliary motion representation. Instead, we directly use a distorted version of the driving image itself to control the transformation.

**Subject-driven Image Generation.** IP-Adapter-FaceID (Ye et al., 2023) and Instant-ID (Wang et al., 2024) extend diffusion-based text-to-image models for subject-driven generation by conditioning on face identity embeddings extracted from external face encoders. Although effective for tasks such as identity interpolation or novel view synthesis, these methods often fail to reproduce the precise pose or expression of a driving image and may introduce unwanted background changes—limiting their applicability to our use case.

## 3 METHOD

### 3.1 LEARNING TO REPLACE FACES IN PHOTOGRAPHS

**Motivation.** Our goal is to train a model that can replace a given source face with a driving face of the *same* individual, potentially captured under different conditions, while preserving the original background. Ideally, this task would require {source, driving, target} triplets, where the target image reflects the desired output. However, to the best of our knowledge, no such dataset exists.

Instead, face reenactment models are typically trained on {source, target} pairs extracted from video frames, where the target image also implicitly serves as the driving signal. Using the full target image directly as input would amount to giving the model access to the ground truth, leading to trivial solutions where the model simply reproduces the target image verbatim. To avoid this, existing

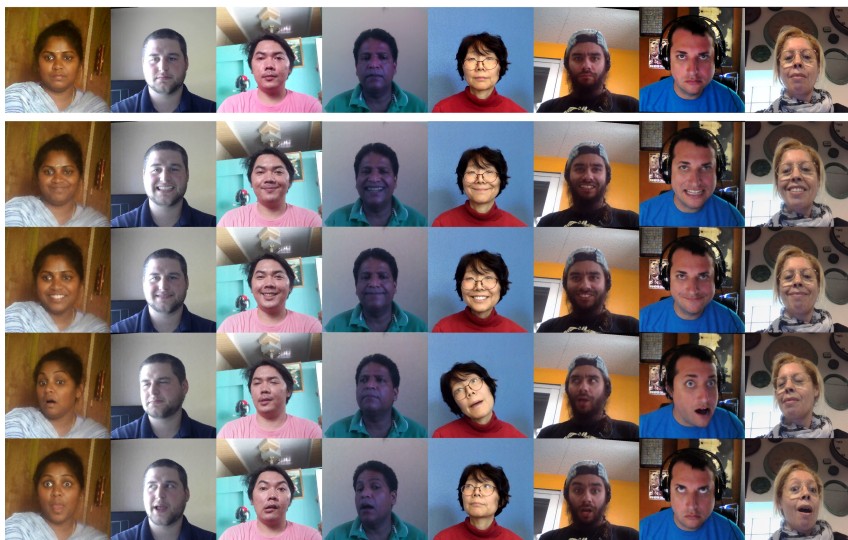

Figure 3: PiQPerfect dataset samples. For each neutral face (top row), we generate 20 expressive variants using a face animation model (Guo et al., 2024). For visualization purposes, we display four randomly sampled expressive variants per neutral face.

methods rely on intermediate representations derived from the target, such as keypoints or learned embeddings. These representations intentionally discard most of the low-level information present in the driving image to prevent trivial copying and allow generalization to driving images depicting different individuals or captured under different conditions.

While such representations are necessary for cross-identity reenactment, they are less suitable for same-identity scenarios. As a result, current face reenactment models must rely heavily on the source image to preserve identity and frequently fail to generate realistic, identity-consistent results that faithfully match the driving image.

**Proposed Representation for the Driving Image.** To address this limitation, we propose using a distorted version of the inner face region from the target image as the driving signal. This simple yet effective strategy provides strong supervision for preserving the facial attributes of the driving image while preventing the model from trivially copying the target.

We begin by segmenting the inner face to exclude the background, ensuring that the model does not rely on the driving signal for this part. This is particularly important for enabling generalization to driving images from external sources, which may contain entirely different backgrounds. We then apply a series of augmentations to the segmented face, including color jittering, grayscale conversion, horizontal flipping, and affine transformations (e.g., translation, rotation, and scaling). Additionally, we leverage a relighting module to introduce artificial shadows and lighting variations. These augmentations help the model generalize to driving images captured under diverse illumination conditions. We then feed our model the obtained distorted face alongside the source image. The model is then trained to reconstruct the original target image from a degraded version of its inner face (and the source image), effectively learning to reverse the applied augmentations. Further details on the data augmentation pipeline are provided in Section 4.

**Model Architecture.** Our diffusion model, illustrated in Figure 2, is based on SDXL (Podell et al., 2024). To condition the model on both the source and driving images, we concatenate the source image, the distorted driving image, and the noisy latent along the width axis before feeding them into the model, taking inspiration from (Shi et al., 2023; Mercier et al., 2024). We experiment with alternative conditioning strategies, including channel-wise concatenation and ControlNet-style (Zhang et al., 2023b) adapter modules, but found spatial concatenation to yield the best performance for this task.

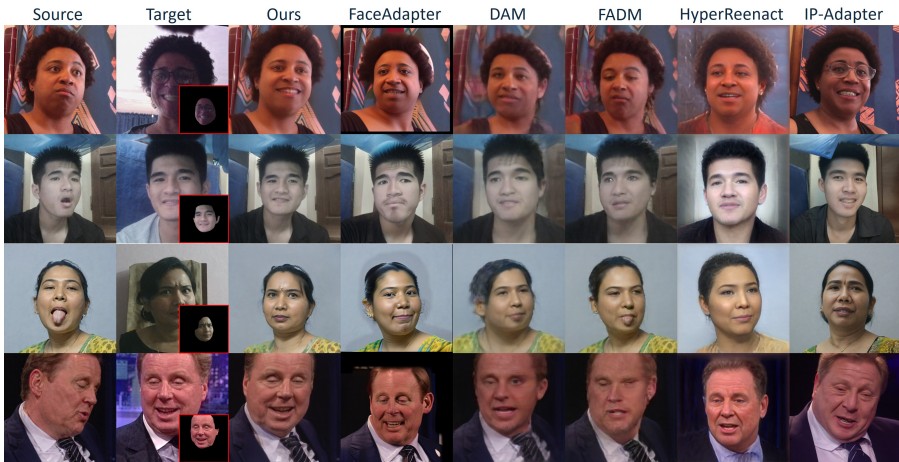

Figure 4: Face Replacement (*in-distribution*). Our method produces high-quality, realistic reconstructions with stronger identity preservation by leveraging both the source and masked driving image (red inset), outperforming other approaches for diverse identities.

Figure 5: Face Replacement (*out-of-distribution*). Our method demonstrates strong generalization for a wide range of facial poses, expressions, identities, and lighting conditions. Moreover, it effectively generates reenacted source images that preserve the subject's identity.

## 3.2 THE PIQPERFECT DATASET

**Existing datasets.** Most face reenactment models are trained using video-based face datasets, where two frames are randomly sampled to serve as source and target images. A common choice are the VoxCeleb datasets (VoxCeleb1 (Nagrani et al., 2017) and VoxCeleb2 (Chung et al., 2018)), which consist of interview videos crawled from YouTube. However, VoxCeleb was originally created for speech-related tasks, where visual fidelity is not a priority. As a result, a significant portion of the VoxCeleb videos are low-resolution, and most clips depict individuals *talking*, with minimal variation in pose or expression throughout. Consequently, randomly sampling two frames often results in pairs with similar facial attributes.

Although more recent datasets such as TalkingHead-1KH (Wang et al., 2021), VFHQ (Xie et al., 2022) and CelebV-HQ (Zhu et al., 2022) offer higher visual quality, they remain heavily biased toward "talking heads" footage crawled from YouTube. Other datasets, such as MEAD (Wang et al., 2020), FaceForensics (Rössler et al., 2018), and UvA-NEMO (Dibeklioğlu et al., 2012), are typically reserved for evaluation due to their smaller size or lack of identity diversity.

**Proposed dataset.** To address these limitations, we introduce the PiQPerfect dataset—a large-scale collection comprising approximately 500,000 images spanning over 6,724 unique identities,

| Methods | Replacement in-distribution (ID) | | | | | | | Replacement out-of-distribution (OOD) | | | | |
|---|---|---|---|---|---|---|---|---|---|---|---|---|
| | CSIM ↑ | FID ↓ | AED ↓ | APD ↓ | AGD ↓ | LPIPS ↓ | PSNR ↑ | CSIM ↑ | FID ↓ | AED ↓ | APD ↓ | AGD ↓ |
| DAM | 0.500 | 24.86 | 2.90 | 0.2604 | 12.45 | **0.0362** | **21.66** | 0.298 | 46.84 | 2.76 | 0.2347 | 15.44 |
| FADM | 0.477 | 48.42 | 2.98 | 0.2643 | 13.41 | 0.0455 | 20.09 | 0.281 | 68.37 | 2.83 | 0.2371 | 16.48 |
| IP-Adapter | 0.282 | 19.28 | 3.07 | 0.2804 | 18.19 | 0.0886 | 15.40 | 0.244 | 31.42 | 2.80 | 0.2713 | 19.86 |
| FaceAdapter | 0.369 | 131.52⋆ | 2.99 | 0.2636 | 11.64 | 0.1127 | 10.43 | 0.282 | 116.62⋆ | 2.80 | 0.2590 | 12.96 |
| HyperReenact | 0.405 | 116.62⋆ | 3.04 | 0.2914 | 11.22 | 0.1038 | 13.69 | 0.317 | 97.93⋆ | 2.80 | 0.2617 | 11.75 |
| LivePortrait | 0.558 | 89.31⋆ | **2.48** | 0.2740 | 8.49 | 0.0391 | 20.30 | - | - | - | - | - |
| Ours | **0.818** | **7.81** | 2.81 | **0.2592** | **5.95** | 0.0640 | 19.17 | **0.605** | **20.26** | **2.64** | **0.2314** | **11.33** |

Table 1: We quantitatively compare our method with prior approaches. We report cosine similarity of ArcFace (Deng et al., 2022) identity embeddings (CSIM), FID, the average distance error for pose (APD), expression (AED), and gaze coefficients (AGD), LPIPS (Zhang et al., 2018), and PSNR. The best scores are highlighted in **blue**, while the second-best scores are underlined. Note: Baselines that failed on a significant number of images leading to unreliable FID scores are indicated by (⋆).

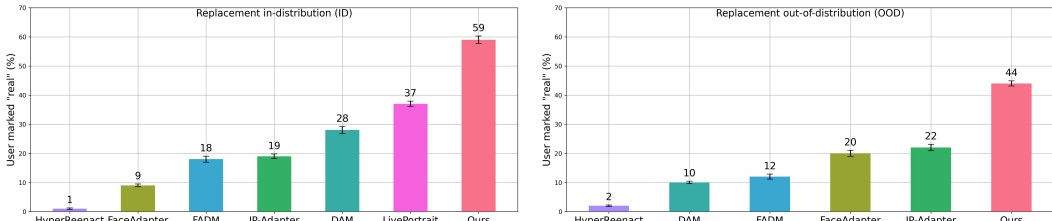

Figure 6: User study: Replacement ID and OOD. Participants were asked to mark {source, output} pairs as "real" or "fake" based on perceptual quality. We report the proportion of images marked as "real" for each approach. The error bars represent the standard error of the mean (SEM).

and from which more than 10 million distinct source–target pairs can be sampled. Sample images from our dataset are shown in Figure 3. Each face in our dataset is available under 21 distinct poses and expressions, including a neutral image (1st row) and 20 expressive variants generated using a face animation model (rows 2-5). To construct the dataset, we leverage an internal face video dataset and LivePortrait (Guo et al., 2024), a model capable of animating a face based on a driving video. We begin by extracting 25,000 neutral faces from our internal dataset, each depicting a person facing the camera with a neutral expression and direct eye contact. We then manually select 200 neutral-to-expressive driving videos, chosen to cover a wide range of facial expressions and head poses. Each neutral face is animated 20 times, each time using a randomly selected driving video, and we retain only the final frame of each animation. This yields 20 expressive images sharing the same background and visual identity as the neutral source.

All images are standardized to high resolution ($1024 \times 1024$). We further generate descriptive captions for each image using LLaVA v1.5–7B (Liu et al., 2024). During training, we randomly sample two images from these sets of 21 (1 neutral + 20 expressive) to form source–target pairs. From each set of 21 images, up to 420 unique source–target combinations can be sampled (excluding self-pairs). With 25,000 such sets, this results in over 10 million unique source–target pairs available for training.

Our dataset captures broad diversity across gender identities, skin tones, and environmental conditions—from dimly lit settings to brightly illuminated scenes, and from close-ups to wide shots. Examples of driving videos used in animation are illustrated in Figure 20 in the appendix. These include a variety of expressions (e.g., *"make a funny face"*, *"smile with mouth opened"*, *"stick your tongue out"*, *"make a sad face"*) and head movements (e.g., *"look left"*, *"tilt your head upward"*).

Compared to sampling two random frames from the same video, our synthetic data generation pipeline allows finer control over dataset composition. For instance, we deliberately oversample driving videos featuring subtle variations of smiling, resulting in expressive faces that appear to be "posing for the camera"—a scenario that is particularly useful for photographic applications such as replacing an unflattering expression in a group photo with a more favorable one from another

image. Such "posed" expressions are rare in existing video-based face datasets, which are typically crawled from YouTube interviews and dominated by conversational or neutral expressions. Another advantage is that, during training, source–target pairs can be selected from dissimilar driving videos to increase task difficulty and model robustness.

## 4 EXPERIMENTS

### 4.1 IMPLEMENTATION DETAILS

**Training Hyperparameters.** We fine-tune the pre-trained SDXL model (Podell et al., 2024) for 25,000 iterations on our proposed dataset (Section 3.2). Training is performed on 8 NVIDIA A100 GPUs with an effective batch size of 64. We use the Adam optimizer with a learning rate of $1 \times 10^{-5}$, $\beta_1 = 0.9$, and $\beta_2 = 0.999$.

**Data augmentation.** Before fine-tuning, we extract segmentation masks for the target image using SegFormer (Xie et al., 2021). During training, these masks are used to isolate the inner face region and exclude the background. To improve robustness, we randomly include optional regions such as hair and neck in the mask. The resulting mask is further refined by applying morphological erosion with a kernel size randomly sampled from the range $[0, 30]$, helping to remove background spillover near the face boundary. To make the model robust to varying lighting conditions, we relight the inner face using NeuralGaffer (Jin et al., 2024), with environment lighting sourced from the PolyHaven dataset (PolyHaven, 2025). The relit inner face is then subjected to a series of augmentations. Finally, the source image, the augmented inner face, and the target image are resized to $576 \times 576$ and encoded using the VAE module (Podell et al., 2024) associated with SDXL. These representations are concatenated along the spatial axis and provided as input to the model. During training, captions are randomly dropped for 10% of the samples to improve robustness to missing text conditioning.

**Evaluation datasets.** Following standard practices in face reenactment, we evaluate our model under two settings: (1) in-distribution (*ID*), where the driving and source images are captured moments apart—for example, in *burst-mode* photo sequences— and (2) out-of-distribution (*OOD*), where the driving image shows the same person under different conditions, for instance when pictures are taken at different times under varying background and lighting conditions. For the *replacement-ID* setting, we use 5,000 test images each from FFHQ and VoxCeleb2. For the *replacement-OOD* setting, we use 5,000 test samples each from PiQPerfect (Section 3.2) and VoxCeleb2.

**Baselines.** We compare our model against five face reenactment approaches: DAM (Tao et al., 2022), FADM (Zeng et al., 2023), Face-Adapter (Han et al., 2024), HyperReenact (Bounareli et al., 2023), and LivePortrait (Guo et al., 2024), as well as one subject-driven image generation baseline, based on IP-Adapter (Ye et al., 2023). LivePortrait is excluded from the OOD setting due to the absence of driving videos compatible with this setup. In the ID setting, we simulate a two-frame video by treating the source and driving images as sequential frames. For the IP-Adapter baseline, we use a dual-adapter SDXL configuration, with each adapter conditioned on a CLIP embedding of the source image and driving image respectively (more details are available in the appendix). For the other baselines, we use the official implementations provided by the authors.

### 4.2 RESULTS

**Replacement in-distribution (ID).** Figure 4 shows a qualitative comparison of replacement-ID results across all methods. Our approach achieves higher reconstruction quality in most examples, producing identity-consistent outputs. In contrast, methods such as DAM, FADM, Face-Adapter, HyperReenact, and LivePortrait primarily rely on the source image for identity, resulting in poor reconstruction quality when the source is in a non-frontal pose (e.g., rows 1 and 3). IP-Adapter, meanwhile, struggles to produce realistic outputs that accurately match the driving image and often fails to preserve the original background.

Table 1 reports quantitative comparisons of replacement-ID performance. Our method outperforms prior approaches on most metrics, with particularly strong results in CSIM, FID, and AGD. Although

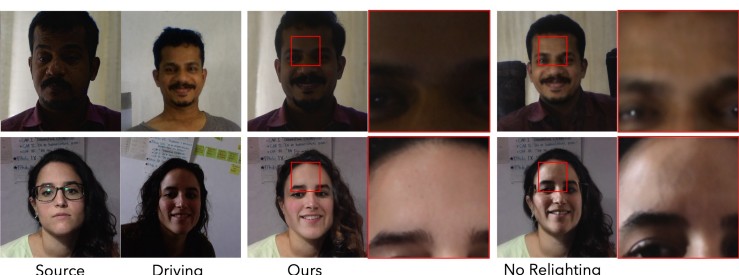

|  | Source | Driving | Ours | Channel Concat | SD v1.5 | No Relighting | No Augmentation | Driving as is |

Figure 7: Ablation study. We evaluate replacing spatial to channel-wise concatenation, swapping SDXL to SDv1.5, as well as disabling relighting, all data augmentations and inner face extraction.

|  | Source | Driving | Ours | | No Relighting | |

Figure 8: Ablation study: Effects of Relighting. Without relighting, lighting artifacts from the driving image often leak into the output. Incorporating random relighting into the data augmentation pipeline produces significantly more realistic and consistent illumination.

| Methods | Replacement in-distribution (ID) | | | | | | | Replacement out-of-distribution (OOD) | | | | |
|---|---|---|---|---|---|---|---|---|---|---|---|---|
|  | CSIM ↑ | FID ↓ | AED ↓ | APD ↓ | AGD ↓ | LPIPS ↓ | PSNR ↑ | CSIM ↑ | FID ↓ | AED ↓ | APD ↓ | AGD ↓ |
| Ours | **0.818** | **7.81** | **2.81** | **0.2592** | **5.95** | 0.0640 | **19.17** | 0.605 | **20.26** | **2.64** | 0.2314 | 11.33 |
| Channel-wise | 0.689 | 55.19 | 3.67 | 0.3542 | 6.18 | 0.0831 | 16.11 | 0.583 | 71.48 | 3.53 | 0.3265 | **10.14** |
| SDv1.5 | 0.776 | 9.90 | 2.87 | 0.2620 | 6.58 | **0.0564** | 17.55 | **0.616** | 21.97 | 2.67 | **0.2288** | 11.45 |

Table 2: Ablation study. We compare spatial versus channel-wise concatenation of the input images and assess the performance with SDv1.5 as the backbone UNet architecture (Rombach et al., 2022).

some baselines achieve better scores on LPIPS and PSNR, these metrics often favor overly smooth or blurry reconstructions over perceptual fidelity (see Figure 12 in the appendix).

**Replacement out-of-distribution (OOD).** A visual comparison of the replacement-OOD results is provided in Figure 5. Compared to prior methods, our approach consistently achieves identity-preserving outputs, even under challenging conditions such as exaggerated expressions in the driving images. Generative baselines like FaceAdapter, IP-Adapter, and HyperReenact struggle with identity preservation and often introduce undesirable alterations to the background of the source image. Landmark-based methods such as DAM and FADM are limited by the low identity information contained in facial landmarks and also face difficulties handling large motion differences between the source and driving images, resulting in blurry or low-quality outputs.

As shown in Table 1, our proposed method quantitatively outperforms all baseline approaches across key evaluation metrics, demonstrating its effectiveness in leveraging facial features from the driving image for accurate face replacement.

**User Study.** As part of our quantitative evaluation, we conducted a user study involving ten participants. Each participant reviewed 25 outputs per model for both ID and OOD face replacement tasks. For each example, participants were shown the source image alongside the model's output and asked to label the result as either "real" or "fake" based on perceptual quality. As shown in

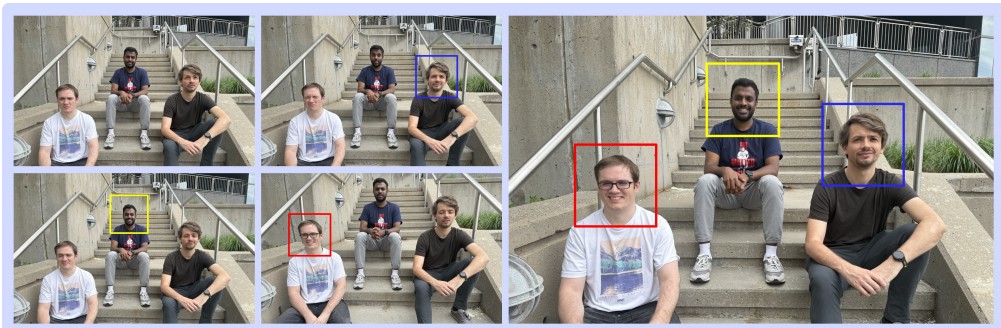

Figure 9: Real-world simulation of composite picture editing ("burst" mode reconstruction)

Figure 6, participants judged our model's outputs to be realistic in 59% of replacement-ID cases and 44% of replacement-OOD cases—substantially outperforming other baselines.

### 4.3 ABLATION STUDY

We conduct a series of ablation studies to evaluate key components of our proposed pipeline. Figure 7 shows visual comparisons for different configurations, and quantitative results for a subset of ablations are reported in Table 2. Our key findings are as follows: Using the target image directly as the driving input results in a degenerate solution in which the model simply reproduces the driving image. Removing all data augmentations severely degrades performance, effectively pasting the driving face onto the source with minimal adjustment in color or structure. Removing relighting introduces "lighting leakage" artifacts—i.e., the model fails to adjust illumination and retains lighting effects from the driving image. Additional results for the no-relighting condition are shown in Figure 8. Channel-wise concatenation leads to consistently poorer performance across all metrics and produces visual artifacts, particularly around fine details such as the teeth. As expected, using SDv1.5 (Rombach et al., 2022) results in slightly lower overall performance. While the SDv1.5-based variant occasionally preserves the pose of the source image more accurately, it tends to produce oversaturated color compositions and lower-quality facial details.

### 5 DISCUSSION AND CONCLUSION

We have introduced *PiQPerfect*, a diffusion-based method for realistic, same-identity facial replacement in photographs. Our approach leverages a pre-trained text-to-image model and a simple but effective training strategy: by feeding the model a distorted version of the target image as the driving signal, we encourage it to rely on the driving image's expression and pose. As a result, the model learns to composite the driving face on the source image while preserving the subject's identity and background. Experimental results demonstrate that our method outperforms existing face reenactment and subject-driven generation techniques on both in- and out-of-distribution face replacement settings. A key part of this work is the PiQPerfect dataset, a large-scale collection of high-quality same-identity image pairs that exhibit diverse facial expressions, poses, and illumination conditions. To support further research, we will release this dataset publicly. A particularly compelling application of our method, illustrated in Figure 9, lies in composite group photo editing, where users can replace unflattering expressions in a group shot with better alternatives from other photos of the same individual, allowing a level of post hoc control difficult to achieve with prior approaches. Although our method produces high-quality and perceptually realistic results, it is not without limitations. Artifacts may appear in fine details such as hair or around the eyes, and performance degrades when the driving face is extremely small or large, likely due to encoding challenges. An interesting future research direction could explore better head pose control by conditioning the model on keypoints or learned facial embeddings. Because our model is trained specifically for same-identity face replacement, it is not designed to handle cross-identity scenarios. Nonetheless, we include qualitative results to illustrate how the model behaves in such cases: some outputs transfer identity, while others transfer only expression (see Figures 16 and 17 in the appendix). These observations highlight interesting directions for future work on disentangling identity and expression.

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

# A APPENDIX

## A.1 ADDITIONAL IMPLEMENTATION DETAILS

**Training.** As part of our data augmentation pipeline, we extract the inner face from driving images and apply relighting. To achieve this, we use SegFormer (Xie et al., 2021) to generate face masks and NeuralGaffer (Jin et al., 2024) to perform relighting, with background lighting sourced from randomly selected images in the PolyHaven dataset (PolyHaven, 2025). To reduce training time and GPU memory usage, both operations are performed offline prior to training. More specifically, for each source image, we generate and save six driving images—five relighted variants and one original—as well as a single face mask. During training, one of the six driving images is randomly sampled for use in each iteration.

**Inference.** Our results are generated using 50 denoising steps and a guidance scale of 1.5, and takes $\sim 3$ seconds per $576 \times 576$ image on a single NVIDIA A100. For classifier-free guidance, we drop the text prompt and use a zeroed-out version of the driving image for the negative branch.

**Baselines.** For most baselines, we use the official implementations provided by the respective authors: DAM[1], FADM[2], HyperReenact[3], Face-Adapter[4], and LivePortrait[5].

For the IP-Adapter baseline, we use the implementation and pre-trained weights available on HuggingFace's diffusers library[6] in a dual-adapter SDXL configuration.[7] One "style" adapter is conditioned on a CLIP embedding of the source image, and the second "face" adapter—fine-tuned for facial representation tasks—is conditioned on a CLIP embedding of the driving image. The "style" and "face" adapters are assigned scale values of 0.5 and 0.6, respectively; these scale parameters control the strength of the corresponding image conditioning. The guidance scale is set to 3.0 to perform 27 denoising steps. In practice, we initialize the process with 30 denoising steps but skip the first three, and initialize the noisy latents with a noisy version of the source image, with the noise strength parameter set to 0.9. For classifier-free guidance, we drop the text prompt and use a noisy version of the driving image as the negative branch.

## A.2 ADDITIONAL RESULTS

**Comparison against Text-driven Image Editing Approaches.** Diffusion models have gained significant attention for instruction-based image editing. SDEdit (Meng et al., 2022) proposes a training-free method for global, text-driven modifications but often fails to preserve regions outside the editing target. Subsequent approaches (Avrahami et al., 2023; Nichol et al., 2021) introduce user-defined masks to enable localized control. Other methods (Cao et al., 2023; Mokady et al., 2023; Parmar et al., 2023) focus on manipulating attention maps, while LEDITS++ (Brack et al., 2024) integrates both attention and noise-based masking to achieve more precise edits. Fully supervised pipelines (Brooks et al., 2023; Zhang et al., 2023a; 2024; Wei et al., 2024a; Zhao et al., 2024) support end-to-end editing from textual instructions. However, these models are not explicitly optimized for identity preservation in face-specific tasks. This limitation is evident in Figure 10, where both InstructPix2Pix and LEDITS++ produce suboptimal results in terms of visual fidelity and identity consistency.

**Quantitative Results.** Tables 3 and 4 provide a per-dataset breakdown of metrics for all models and each task. Please note that the FID scores for Face-Adapter (Han et al., 2024) and Hyper-Reenact (Bounareli et al., 2023) may be unreliable due to face detection failures during evaluation. Specifically, Face-Adapter fails on 4,325 (FFHQ), 4,813 (VoxCeleb ID), 68 (PiQPerfect), and 4,896 (VoxCeleb OOD) images out of 5,000. HyperReenact fails on 297 (FFHQ), 1,052 (VoxCeleb ID),

---

[1] https://github.com/JialeTao/DAM
[2] https://github.com/zengbohan0217/FADM
[3] https://github.com/StelaBou/HyperReenact
[4] https://github.com/FaceAdapter/Face-Adapter
[5] https://github.com/KwaiVGI/LivePortrait
[6] https://huggingface.co/docs/diffusers/using-diffusers/ip_adapter
[7] The adapter names are *ip-adapter-plus_sdxl_vit-h"* and *ip-adapter-plus-face_sdxl_vit-h"*

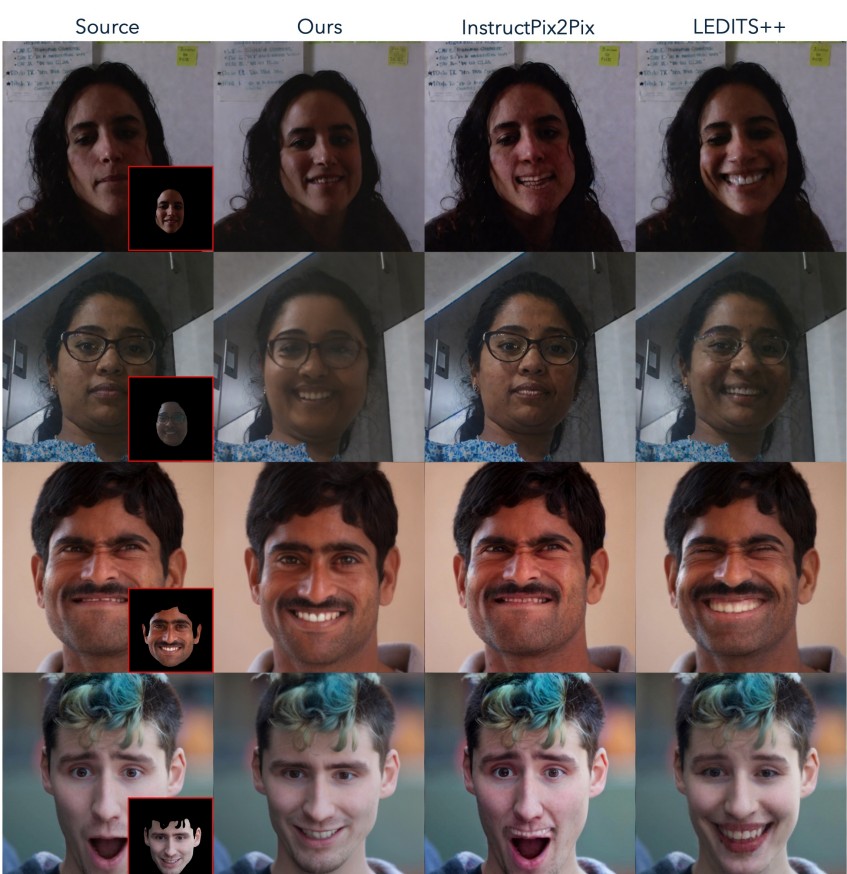

Figure 10: Comparing our facial replacement with instruction-based image editing approaches (Brooks et al., 2023; Brack et al., 2024). Note that, our approach uses a masked driving image (show in red inset), while the image editing approaches do not require a driving image. Our approach makes realistic edits following the driving image, while the image editing methods have no control over the facial edits, leading to unrealistic changes to personal identity.

901 (PiQPerfect), and 1,264 (VoxCeleb OOD) images. These failures may significantly impact the quality of FID estimation for these methods.

**Issues of image quality metrics.**    In Figure 12, we highlight the discrepancy between standard image quality metrics and human judgment. Although our results exhibit superior visual quality compared to baseline methods—as further supported by our user study—the quantitative metrics tend to undervalue our approach.

**Model robustness to various transformations.**    In Figure 13, we assess our model's robustness to various augmentations applied to the face in the driving image, including color jittering, translation, rotation, and scaling. The results indicate that our model is generally consistent despite large variations in facial color, position, orientation, and size. However, it exhibits difficulty in preserving identity when the face in the driving image is either extremely small (first row of Scaling driving) or excessively large (fourth row of Scaling driving). This limitation may stem from information loss during the image-to-latent conversion process—where small faces lack sufficient detail.

**Additional visual results.**    Figures 14 and 15 present additional visual results of our proposed method for the face replacement task.

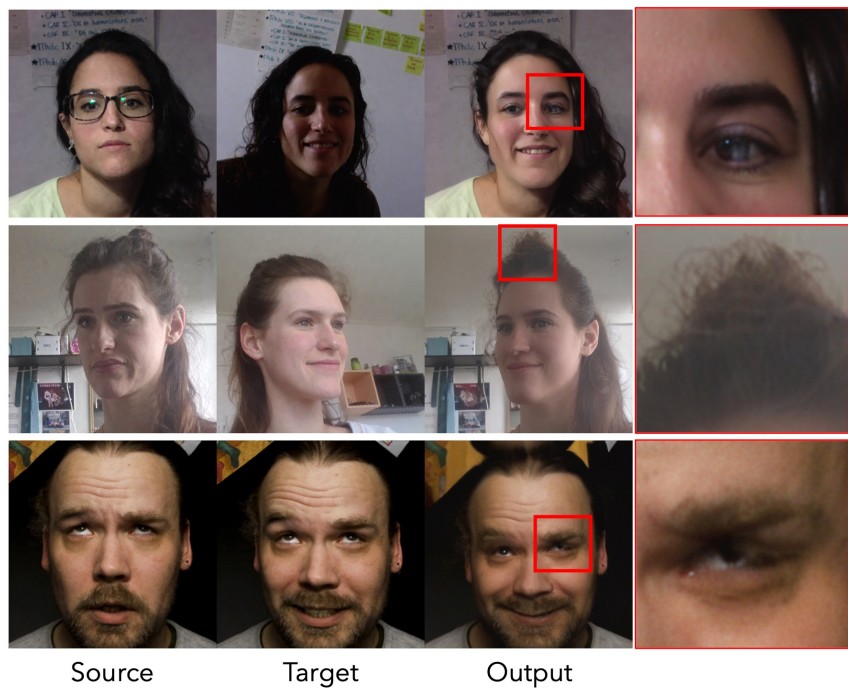

Source       Target       Output

Figure 11: Although our proposed approach achieves more realistic facial replacement compared to prior methods, it still exhibits limitations in certain areas, particularly around the eyes and hair.

| | | | | | | | |
|---|---|---|---|---|---|---|---|
| | | | | **Replacement in-distribution (ID)** | | | |
| Datasets | Methods | CSIM ↑ | FID ↓ | AED ↓ | APD ↓ | AGD ↓ | LPIPS ↓ | PSNR ↑ |
| FFHQ | DAM | 0.557 ± 0.0034 | 25.28 | 2.91 ± 0.0128 | 0.2796 ± 0.0022 | 11.89 ± 0.1247 | **0.0380 ± 0.0003** | **21.37 ± 0.0491** |
| | FADM | 0.542 ± 0.0034 | 43.24 | 2.97 ± 0.0138 | 0.2849 ± 0.0022 | 13.28 ± 0.1278 | 0.0453 ± 0.0004 | 19.76 ± 0.0464 |
| | IP-Adapter | 0.312 ± 0.0018 | 17.22 | 3.01 ± 0.0116 | 0.2910 ± 0.0022 | 16.28 ± 0.1839 | 0.0835 ± 0.0004 | 14.78 ± 0.0312 |
| | FaceAdapter | 0.429 ± 0.0064 | 122.47 | 2.97 ± 0.0327 | 0.2439 ± 0.0049 | 10.37 ± 0.3383 | 0.1252 ± 0.0009 | 10.32 ± 0.0824 |
| | HyperReenact | 0.431 ± 0.0019 | 111.89 | 2.79 ± 0.0118 | **0.2439 ± 0.0019** | 11.14 ± 0.0914 | 0.1152 ± 0.0004 | 13.56 ± 0.0245 |
| | LivePortrait | 0.676 ± 0.0031 | 11.18 | **2.79 ± 0.0113** | 0.2804 ± 0.0021 | 8.44 ± 0.0903 | 0.0432 ± 0.0003 | 20.23 ± 0.0584 |
| | Ours | **0.863 ± 0.0017** | **6.51** | 2.86 ± 0.0116 | 0.2797 ± 0.0021 | **5.35 ± 0.0647** | 0.0620 ± 0.0003 | 18.46 ± 0.0434 |
| VoxCeleb | DAM | 0.466 ± 0.0031 | 24.43 | 2.89 ± 0.0138 | 0.2412 ± 0.0017 | 13.01 ± 0.1220 | 0.0345 ± 0.0003 | **21.96 ± 0.0433** |
| | FADM | 0.413 ± 0.0032 | 53.60 | 2.99 ± 0.0163 | 0.2437 ± 0.0017 | 13.55 ± 0.1268 | 0.0457 ± 0.0003 | 20.43 ± 0.0436 |
| | IP-Adapter | 0.253 ± 0.0018 | 21.33 | 3.12 ± 0.0133 | 0.2698 ± 0.0019 | 20.10 ± 0.2033 | 0.0938 ± 0.0003 | 16.03 ± 0.0279 |
| | FaceAdapter | 0.309 ± 0.0126 | 140.57 | 3.14 ± 0.0658 | 0.2833 ± 0.0086 | 12.90 ± 0.7425 | 0.1003 ± 0.0027 | 10.54 ± 0.1666 |
| | HyperReenact | 0.379 ± 0.0022 | 100.97 | 3.29 ± 0.0141 | 0.3147 ± 0.0018 | 11.29 ± 0.1034 | 0.0925 ± 0.0004 | 13.82 ± 0.0254 |
| | LivePortrait | 0.486 ± 0.0029 | 13.57 | **2.72 ± 0.0116** | 0.2455 ± 0.0017 | 10.37 ± 0.1037 | **0.0290 ± 0.0002** | 20.78 ± 0.0517 |
| | Ours | **0.772 ± 0.0027** | **9.12** | 2.76 ± 0.0122 | **0.2388 ± 0.0017** | **6.55 ± 0.0762** | 0.0660 ± 0.0003 | 19.89 ± 0.0389 |

Table 3: We quantitatively compare our method with prior approaches (± Standard Error of Mean) on replacement-ID testsets FFHQ (Karras et al., 2019) and VoxCeleb (Nagrani et al., 2017; Chung et al., 2018).

**Failure cases.** Figure 11 illustrates representative failure cases in the same-identity face replacement task. Most failures are caused by suboptimal generation of intricate details such as eyes and hair, and this is caused by the limitations of the backbone diffusion (Podell et al., 2024) architecture.

| | | Replacement out-of-distribution (OOD) | | | | | | |
|---|---|---|---|---|---|---|---|---|
| Datasets | Methods | CSIM ↑ | FID ↓ | AED ↓ | APD ↓ | AGD ↓ | LPIPS ↓ | PSNR ↑ |
| PiQPerfect | DAM | 0.350 ± 0.0028 | 55.16 | 2.61 ± 0.0096 | 0.2222 ± 0.0017 | 15.29 ± 0.1400 | **0.1087 ± 0.0008** | **14.39 ± 0.0552** |
| | FADM | 0.342 ± 0.0027 | 74.98 | 2.66 ± 0.0102 | 0.2252 ± 0.0018 | 16.51 ± 0.1498 | 0.1112 ± 0.0006 | 13.66 ± 0.0499 |
| | IP-Adapter | 0.275 ± 0.0017 | 39.16 | 2.45 ± 0.0075 | 0.2666 ± 0.0024 | 16.59 ± 0.1881 | 0.1375 ± 0.0005 | 12.00 ± 0.0350 |
| | FaceAdapter | 0.343 ± 0.0023 | 75.37 | 2.53 ± 0.0077 | 0.2076 ± 0.0016 | 11.31 ± 0.1003 | 0.1251 ± 0.0008 | 10.99 ± 0.0361 |
| | HyperReenact | 0.359 ± 0.0021 | 92.31 | **2.25 ± 0.0071** | **0.1999 ± 0.0018** | 11.52 ± 0.0966 | 0.1617 ± 0.0007 | 11.47 ± 0.0328 |
| | Ours | **0.631 ± 0.0029** | **28.08** | 2.50 ± 0.0079 | 0.2262 ± 0.0017 | **9.87 ± 0.1317** | 0.1151 ± 0.0007 | 12.27 ± 0.0396 |
| VoxCeleb | DAM | 0.246 ± 0.0021 | 38.51 | 2.91 ± 0.0135 | 0.2472 ± 0.0017 | 15.59 ± 0.1355 | 0.1248 ± 0.0007 | **13.05 ± 0.0388** |
| | FADM | 0.219 ± 0.0022 | 61.76 | 3.01 ± 0.0163 | 0.2490 ± 0.0018 | 16.45 ± 0.1473 | 0.1290 ± 0.0005 | 12.54 ± 0.0353 |
| | IP-Adapter | 0.214 ± 0.0016 | 23.68 | 3.14 ± 0.0134 | 0.2761 ± 0.0020 | 23.13 ± 0.2305 | 0.1301 ± 0.0004 | 11.92 ± 0.0286 |
| | FaceAdapter | 0.221 ± 0.0119 | 157.87 | 3.06 ± 0.0814 | 0.3104 ± 0.0145 | 14.61 ± 1.1722 | 0.1221 ± 0.0030 | 9.34 ± 0.1830 |
| | HyperReenact | 0.275 ± 0.0021 | 103.55 | 3.34 ± 0.0150 | 0.3235 ± 0.0019 | **11.99 ± 0.1150** | 0.1358 ± 0.0008 | 11.54 ± 0.0328 |
| | Ours | **0.579 ± 0.0030** | **12.44** | **2.78 ± 0.0122** | **0.2366 ± 0.0017** | 12.78 ± 0.1606 | **0.1196 ± 0.0006** | 12.02 ± 0.0309 |

Table 4: We quantitatively compare our method with prior approaches (± Standard Error of Mean) on replacement-OOD testsets PiQPerfect and VoxCeleb (Nagrani et al., 2017; Chung et al., 2018).

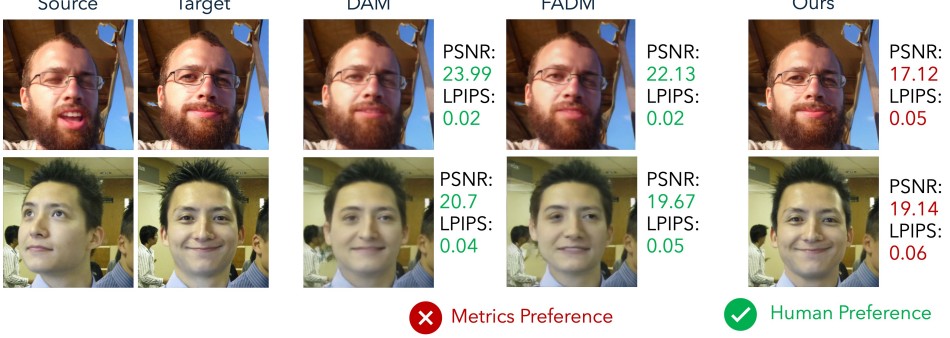

Figure 12: Metrics like PSNR and LPIPS often favor blurry, low-quality reconstructions due to sensitivity to slight misalignments, failing to align with human judgment.

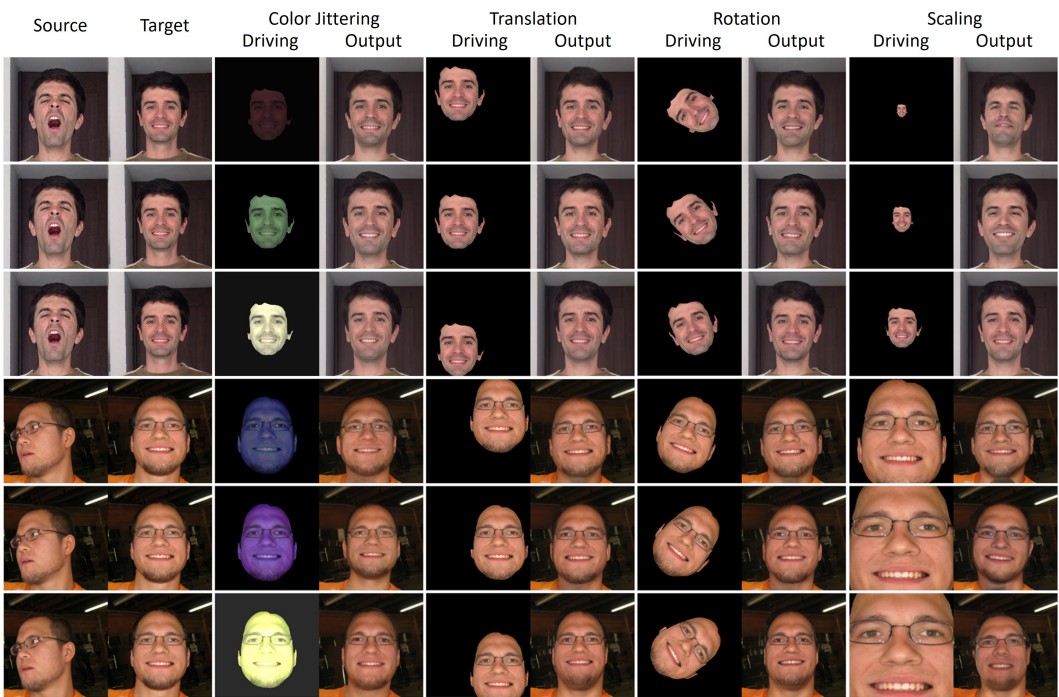

Figure 13: Ablation: Robustness to data augmentations. Our solution demonstrates the ability to mitigate the effects of different data augmentations, resulting in more stable and visually appealing results.

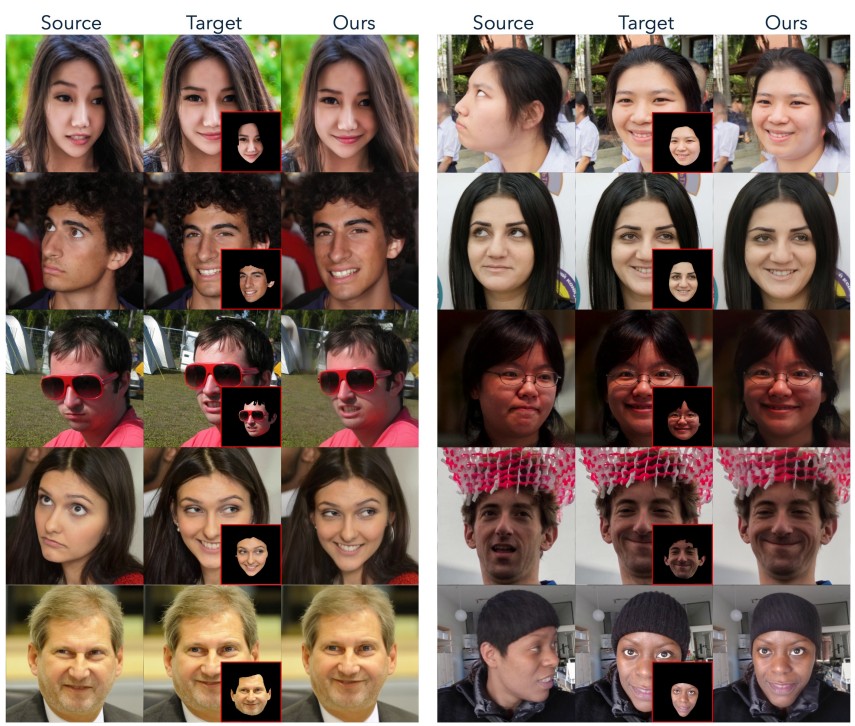

Figure 14: Additional face replacement (in-distribution) results

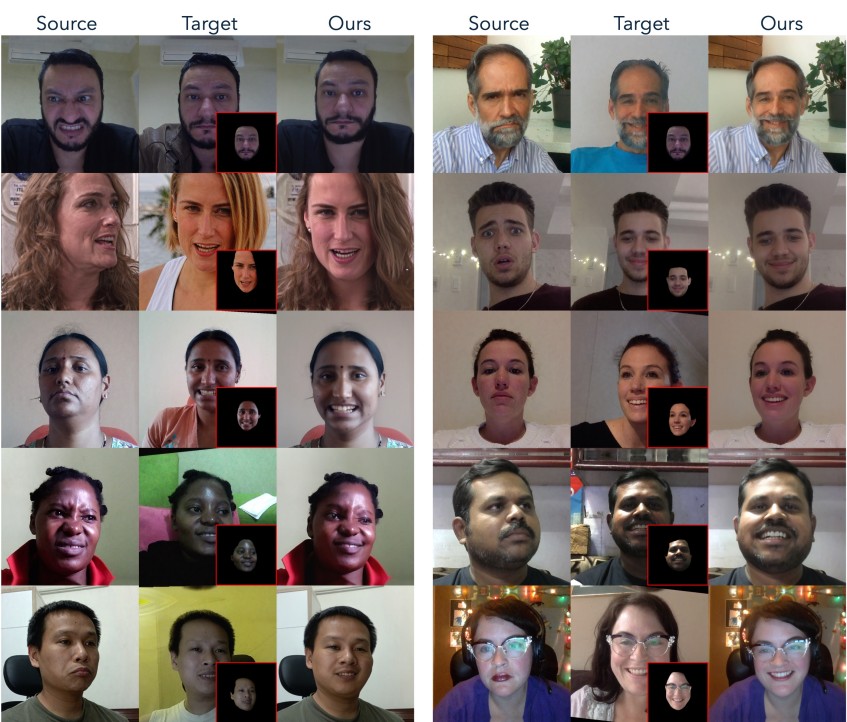

Figure 15: Additional face replacement (out-of-distribution) results

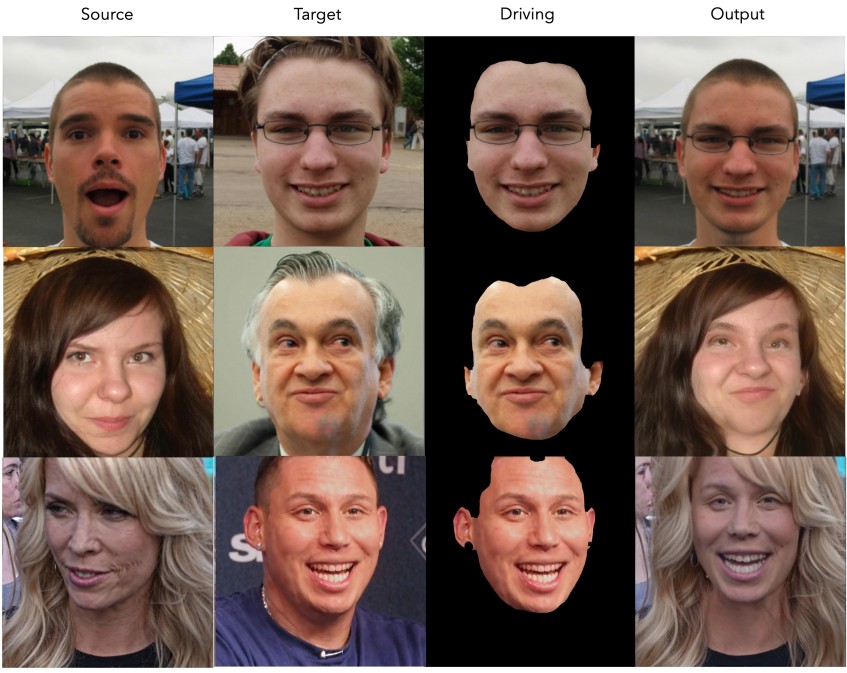

Figure 16: Experiment with cross-identity facial replacement where the identity along with the expression from the driving image gets transferred to source.

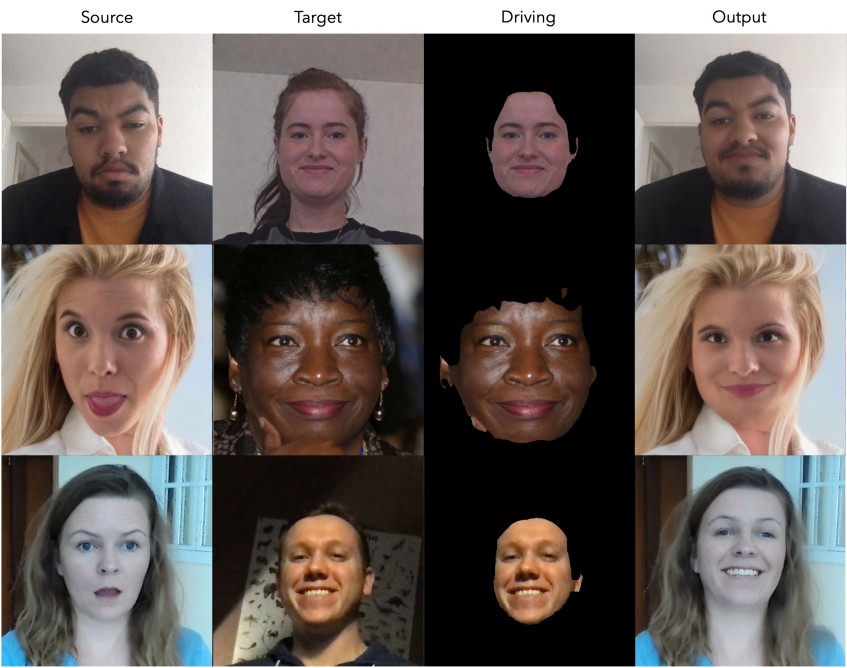

Figure 17: Experiment with cross-identity facial replacement where only the expression from the driving image gets transferred to source while keeping the identity unaltered.

## B  PIQPERFECT DATASET

Most existing reenactment datasets are biased toward interview-style YouTube videos, where subjects are primarily talking with limited head movement and facial variation. In contrast, we introduce the PiQPerfect dataset, a synthetic dataset featuring diverse head poses and expressions, generated using a portrait animation pipeline. Our dataset comprises approximately 500,000 images spanning over 6,724 unique identities, and from which more than 10 million distinct source–target pairs can be sampled. Sample images from our dataset are shown in Figure 18. Each face in our dataset is available under 21 distinct poses and expressions, including a neutral image and 20 expressive variants generated using LivePortrait (Guo et al., 2024), a face animation model.

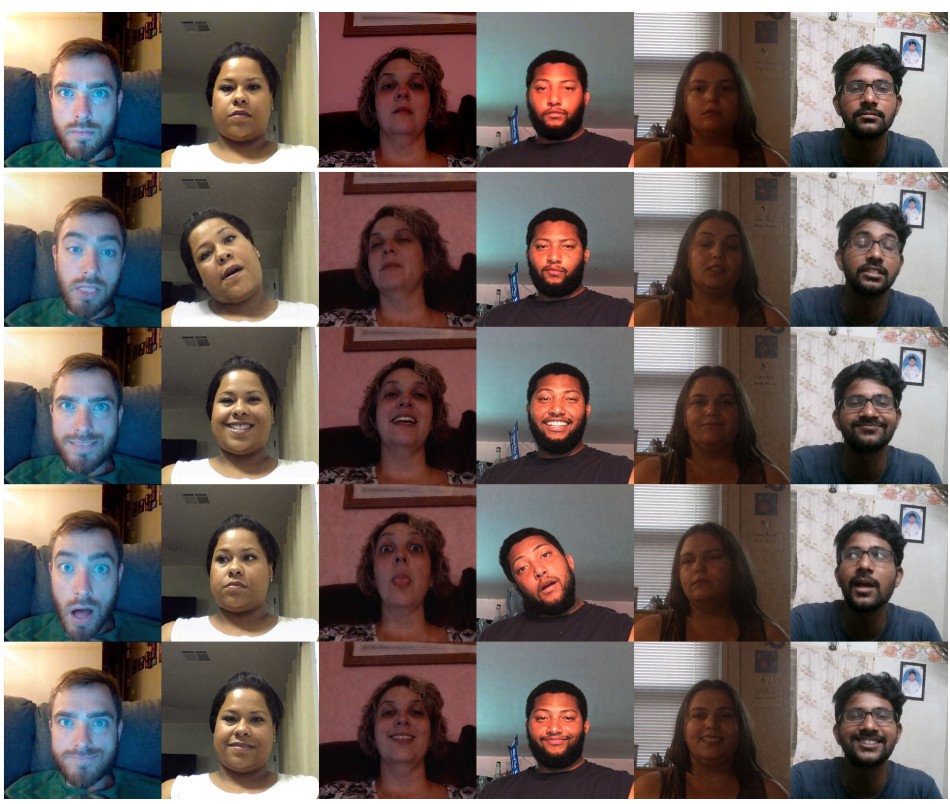

Figure 18: PiQPerfect dataset samples. For each neutral face (top row), we generate 20 expressive variants using a face animation model (Guo et al., 2024). For visualization purposes, we display four randomly sampled expressive variants per neutral face.

To construct the dataset, we leverage an internal face video dataset and LivePortrait (Guo et al., 2024), a model capable of animating a face based on a driving video. An overview of the animation pipeline is shown in Figure 19.

### B.1  VIDEO CROWDSOURCING

To collect videos, we used a custom platform in collaboration with crowdsourcing providers to recruit participants from diverse gender, geographic, and ethnic backgrounds. Participants were instructed to record short videos while performing specific actions and expressions, such as "smiling gently at the camera while sitting in front of a computer", "pretending to be shocked while looking at the screen", "looking down with a sad expression while sitting at a desk" and so on. Each participant was encouraged to submit multiple videos featuring varied backgrounds, outfits, and lighting conditions.

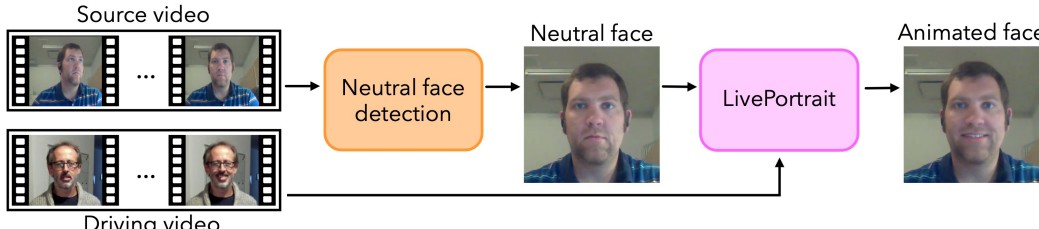

Figure 19: PiQPerfect Dataset generation process overview: Neutral looking faces are detected and extracted from source videos, which are guided via driving videos to generate expressive animated faces using LivePortrait (Guo et al., 2024).

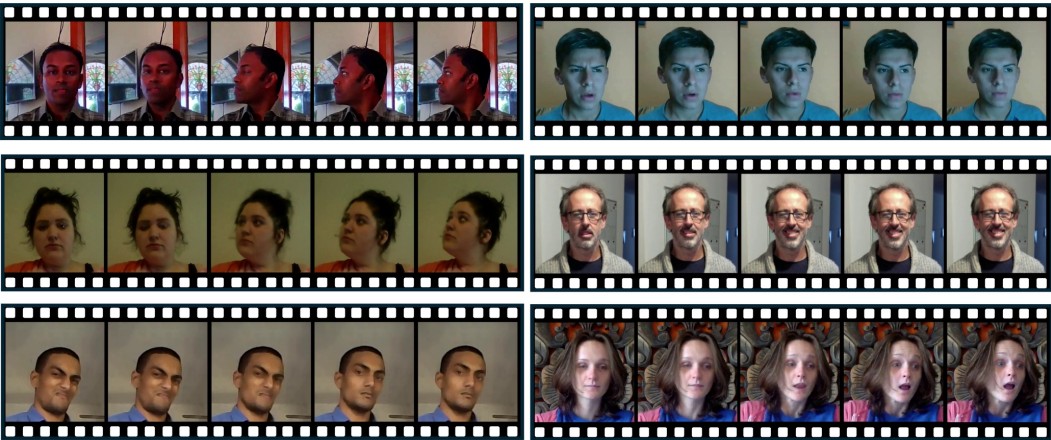

Figure 20: Overview of some example driving videos used to animate the neutral faces with Live-Portrait (Guo et al., 2024).

## B.2 DATA PRE-PROCESSING AND FILTERING

From the collected videos, we filter and retain only those that fall into the following predefined categories of facial expressions and head gestures:

- making eye contact with the camera while sitting in front of a computer
- making eye contact with the camera while sitting at a distance
- making eye contact with the camera while standing
- no facial expression
- no gesture

From the filtered videos, we randomly sample pairs of frames from segments where the subject is looking directly at the camera. To ensure the selection of truly neutral expressions, we extract facial landmarks and bounding boxes using FaceNet (Esler). We then compute the angles between the left and right eyes and the nose, as well as left and right corners of the mouth and the nose, ensuring the head orientation is within an empirically determined threshold of $10°$. To further eliminate head tilt, we also check that the vertical angle between the nose and the mid-point of both the eyes is below $10°$.

Images that pass both filters are standardized by cropping around the midpoint of the detected bounding box, using a bounding box scaled by a factor of 2. This process yields approximately 25,000 neutral face images across 6,724 unique identities.

To enable neutral-to-any-expression animation, we manually select 200 diverse driving videos. In Figure 20, we provide some example driving videos. For each neutral face, we randomly sample 20

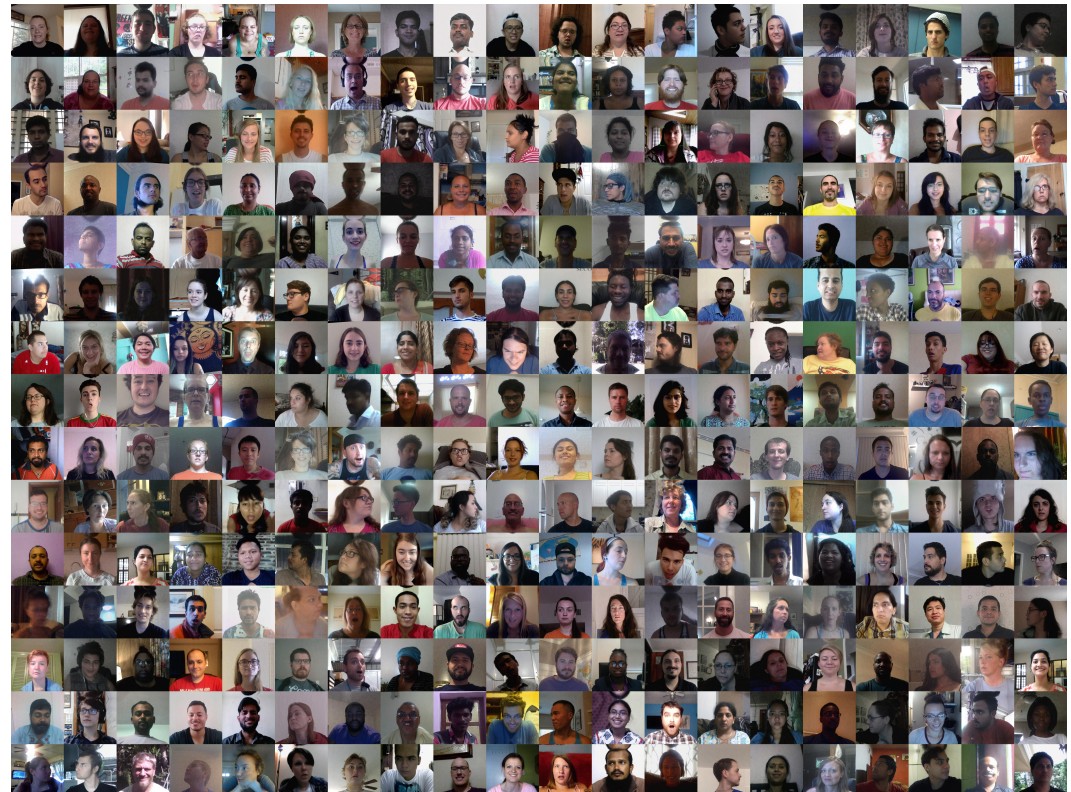

Figure 21: Random samples from the dataset. The PiQPerfect dataset features a wide diversity of backgrounds, lighting conditions, ethnicities, clothing styles, poses, and facial expressions.

unique driving videos and animate the source using LivePortrait (Guo et al., 2024). An overview of the animation pipeline using LivePortrait is shown in Figure 19. From each resulting video, we extract the final frame as the animated output. This results in 20 animated images per source, culminating in a dataset of approximately 500,000 images comprising both neutral and expressive faces.

We provide an overview of the images in Figure 21 to demonstrate the data diversity for backgrounds, lighting, gender, ethnicity, head poses, facial expression.

### B.3 Caption Generation

To generate descriptive captions for the upscaled images, we use LLaVA-1.5-7B (Liu et al., 2024) with the instruction: "*Describe this real-world image and its style as a long detailed caption. Include details about the person's facial expression and describe face attributes. Ignore the background and clothing information for the caption.*"

### B.4 Potential Social Impact

We present a dataset for realistic, same-identity face replacement, aimed at positive applications such as composite group photo editing. However, like other generative technologies, models trained on our dataset carry risks of misuse, such as identity theft and misinformation. We strongly condemn and oppose any deceptive or harmful use of our system. Although our experiments show that it is possible to obtain visually compelling results, the outputs still contain detectable artifacts, as highlighted in the failure cases as shown in Figure 11. To mitigate potential misuse, we propose the embedding of digital watermarks to verify the authenticity and provenance of the image. Additionally, our dataset could be used to train forgery detection models capable of identifying tampered content.

