# OpenReview forum: "PiQPerfect: Diffusion-based Same-identity Facial Replacement"
_ICLR.cc/2026/Conference — ICLR 2026 Conference Withdrawn Submission_

### Official Review · Reviewer_LzSH · 2025-10-31

**Soundness:** 2
**Presentation:** 3
**Contribution:** 2
**Rating:** 4
**Confidence:** 5

**Summary:**

This paper proposes PiQPerfect, a diffusion-based method for same-identity facial replacement. Built on a pre-trained text-to-image model (SDXL), it replaces a face in a source image with the expression and pose from a reference image of the same person, preserving identity and background. Unlike prior cross-identity face reenactment methods that struggle with identity preservation, PiQPerfect leverages a distorted version of the driving face as input to avoid trivial copying and enhance realism. The authors also introduce the PiQPerfect dataset, a large-scale collection of same-identity image pairs with diverse expressions and poses. Experiments show superior performance over existing methods in both in-distribution (ID) and out-of-distribution (OOD) settings.
This paper also introduce a PiQPerfect dataset (~500k images, 6,724 identities) synthesized by animating ~25k neutral faces with LivePortrait to produce 20 expressive variants per identity, yielding >10M source–target pairs.

**Strengths:**

- The paper effectively identifies a practical yet underexplored gap—same-identity facial replacement—and convincingly argues why existing cross-identity reenactment methods are suboptimal for this specific task.
- The use of a distorted, masked driving face as input is an elegant solution that avoids trivial copying while providing strong supervision for pose and expression, directly addressing the core challenge of identity preservation.
- Multiple metrics (CSIM, FID, APD/AED/AGD, LPIPS/PSNR), ID vs OOD settings, and user study; appendix acknowledges baseline failures affecting FID reliability.
- High-Quality, Purpose-Built Dataset: The introduction of the PiQPerfect dataset, with controlled generation of diverse expressions and poses from neutral faces, fills a critical data void and enables robust training for the target application.

**Weaknesses:**

- Missing Direct Comparison to Closest Contemporary Diffusion-Based Work. Despite categorizing the field and comparing to existing face-reenactment baselines, the paper omits discussion and direct comparison with several concurrent diffusion-based face swapping and same-identity replacement works, specifically Baliah et al. (2025)[1], Wang (2024)[2], and Galanakis et al. (2025)[3]. For example, DiffFace and REFace attempt diffusion-based face swapping leveraging facial guidance and are highly pertinent; not benchmarking against them weakens the empirical argument, especially as these methods may target similar strengths (identity, realism).

- The method described in this article was tested on the outdated SDXL and its migration performance was not tested on the modern DiT architecture. Currently, DiT has become mainstream in the field of image editing and generation.

- Artifacts and Failure Cases. The method, while overall strong, is not free from notable artifacts. Figure 11 and supplemental figures clearly display issues around eyes, hair, and illumination blending, especially in hard edge regions and with extreme poses. These failure modes would benefit from a more rigorous analysis—such as error maps or a systematic breakdown—rather than merely attributing them to limitations of the backbone.

Reference:
1.  Baliah, S., Lin, Q., Liao, S. (2025): "REFace: Realistic and Efficient Face Swapping: A Unified Approach with Diffusion Models" WACV2025
2.  Wang, F. (2024): "Face Swap via Diffusion Model"
3.  Galanakis, S., Lattas, A., Moschoglou, S. (2025): "FitDiff: Robust Monocular 3D Facial Shape and Reflectance Estimation using Diffusion Models"

**Questions:**

See  Weakness.

---

> ### Author Response · Authors · 2025-11-14
>
> We thank the reviewer for the feedback and are encouraged by the recognition of our method’s effectiveness, practical relevance, and the contribution of the dataset. We address the main concerns below.
>
> ---
>
> > About *“missing comparisons with closest contemporary diffusion-based work”*
>
> We thank the reviewer for this suggestion and plan to add details about these baselines in the Related Work section.
>
> ---
>
> > About *“tests with the modern DiT architectures”*
>
> We expect the same results can be obtained on DiT architectures as no SDXL-specific techniques were used.

---

### Official Review · Reviewer_PPxW · 2025-10-31

**Soundness:** 2
**Presentation:** 3
**Contribution:** 1
**Rating:** 2
**Confidence:** 4

**Summary:**

The paper presents a diffusion based approach towards same-identity facial replacement in realistic photographs. The method revolves around finetuning a pretrained SDXL for this task on a unique dataset of source, replacement and target images. The primary insight appears to be using augmented copies of the replacement image to avoid trivial copying. The authors also contribute a dataset of triplets created using facial animation models. The paper shows empirical results for VoxCeleb, FFHQ and their new dataset.

**Strengths:**

1. The paper addresses a specific niche task of same face replacement that has not been addressed significantly before. The approach has fairly interesting use cases that may be relevant to modern imaging systems.

2. The authors release a new, well-curated dataset for face replacement that would be a useful training and evaluation benchmark.

3. The presented empirical benchmarks show the models outperform existing face-replacement approaches on a variety of metrics.

**Weaknesses:**

1. The baseline comparisons do not consider models/pipelines that could be used for this task (for example, SimSwap-family, Roop variants, and even opensource/commercial general image generation/editing model ). The paper compares mainly to academic reenactment systems + one IP-Adapter setup (and some editing methods in the appendix) but should at least show the results of modern foundational generative models on this task as a baseline.

2. The generated training dataset relies on synthetic data generated using LivePortrait. However the OOD evaluation also uses a subset of the generated dataset, which overestimates the generalization performance.

3. The draft mentions an internal dataset as being the seed for the generated data. Since this involves human images, I believe a diversity and demographic analysis of the data, and the synthetic data should be included with the paper. Also, the paepr does not mention if the human subjects were asked for consent.

4. The presented approach also does not add any significant novelty beyond the dataset. The stacked reference image approach has been previously used by several state of the art image-gen models (Flux, Seedream).

Overall, I suggest the authors compare with frontier image gen models to see how they stack up on the same task to further bolster their empirical results and novelty of the work.

**Questions:**

See weaknesses above

**Details Of Ethics Concerns:**

The authors mention the use of an internal dataset of human images. Since they do not mention consent or provide any details, I wanted to flag ethics reviewers for a deeper look.

---

> ### Author Response · Authors · 2025-11-14
>
> We thank the reviewer for recognizing the quality of our results and our method’s usefulness for practical expression editing. We also appreciate the acknowledgement of our dataset contribution, which we consider a key part of this work. We address the main concerns below.
>
> ---
>
> > About *“showcasing results of modern foundational generative models as baselines”*
>
> We thank the reviewer for their suggestion. We plan on evaluating these baselines and including comparisons in the supplementary material in a future version of the paper.
>
> ---
>
> > About *“model generalizability on real-world data”*
>
> We acknowledge the importance of this concern and believe that this needs further clarification. Despite the entire training dataset being synthetically generated, **half of the “out-of-distribution” test-set is real-world data**. We followed a different process for the 5,000 PiQPerfect test-images in this setup, where two random frames were captured from two separate videos submitted by the same identity. The results on these test-sets ensure that **the model generalizes well to real-world as well as synthetically generated data**. Additionally, the example shown in Figure 9 showcases another potential real-world application of our solution to composite photograph editing. The 4 photographs on the left half of the figure were taken in a “burst-mode” setup, while the one on the right is the result of applying our model to the faces of the input (top-left image), and using faces from the other pictures (highlighted in colored boxes) as driving images.
>
> ---
>
> > About *“details on the internal video dataset and consent documentation”*
>
> The supplementary material already includes several visualizations that detail the proposed benchmark:
> - Figure 18: Representative samples from the PiQPerfect dataset
> - Figure 19: Overview of the dataset generation pipeline
> - Figure 20: Example driving videos used to animate neutral faces via LivePortrait
>
> The data collection was performed under direct agreements with crowd-workers, permitting both research and commercial use of their video and audio data, including training AI models, and ensuring compliance with privacy regulations. Personally identifiable information and metadata were removed to ensure participant privacy. Contributors received fair and appropriate compensation according to regional standards. A contact email will be provided on the dataset release page for participants to request data removal at any time.

---

### Official Review · Reviewer_aGW9 · 2025-11-02

**Soundness:** 2
**Presentation:** 3
**Contribution:** 2
**Rating:** 4
**Confidence:** 5

**Summary:**

This paper proposes a dataset and a model for replacing the face in the source image with that in the target image. The key is using a driving image capturing just the desired face with everything else blacked out so that the model can be trained without just providing the target image (which usually leads to trivial solutions). The target image can be of a different background or lighting condition, and only the face part should swapping.

The model called PiQPerfect segments out the face from the target image, data augments it, encodes the source, target, driving images, concats them with noise added to the target image (so that the model can’t cheat), and finally diffuses the stack into the target image.

The said dataset contains 500K images of almost 7K identities. Data come in (source, driving, target) tuples. The secret to this big size is that the authors started with 25K neutral faces and then autolabeled many new expressions using LivePortrait.

**Strengths:**

This is indeed a common use case, and I’m also surprised there was no prior work directly tackling this.

Good presentation, esp Figure 2, helped demonstrate the ideas clearly.

I like the autolabeling pipeline that enlarges the dataset by a lot. Also, the paper showed it’s not enough to just use LivePortrait since the background may change.

**Weaknesses:**

The method looks quite similar to many prior face works, one of which is DiffusionRig. DiffusionRig also targets identity preserving editing, but I am not seeing comparisons against DiffusionRig.

The wording on the dataset release sounds vague, so I’m not sure if the authors will eventually release the dataset or not. If the dataset is not released, the only useful part would be the method, but like I said above, it’s not much different than prior works. I’d say the whole package – the model, dataset, and the autolabeling pipeline – is much more valuable than just the model.

**Questions:**

Have you tried this on video? Can be per-frame. How will the temporal consistency look if we make a video of the same person with the face swapped?

Will the dataset be released if this paper gets accepted?

What was done to pseudo GT with artifacts from LivePortrait? Do you have filtering in the data pipeline?

**Details Of Ethics Concerns:**

Potentially a dataset of many faces will be released

---

> ### Author Response · Authors · 2025-11-14
>
> We are encouraged that the reviewer found our task setup clear and relevant, and our solution effective. Below we address the specific concerns raised.
>
> ---
>
> > About *“comparisons against DiffusionRig”*
>
> We thank you for the suggestion. We do not believe that this method can be directly applied to our evaluation setup, however we will mention this in the Related Work section.
>
> ---
>
> > About *“details regarding dataset release”*
>
> Upon acceptance, we plan to release the dataset under a research friendly license. Specifically, we plan to release (neutral + expressive) face images as well as driving videos, in case models other than LivePortrait are preferred.
>
> ---
>
> > About *“application to videos and temporal consistency”*
>
> Our method is trained on static images, and pose is not explicitly controlled. As a result, applying it frame-by-frame to videos does not guarantee temporal consistency.
>
> ---
>
> > About *“filtering pseudo ground-truth images from LivePortrait”*
>
> A simple automated data filtering process was followed to clean the outputs of LivePortrait – ensuring the face is aligned with the center of the image, and that the face is detected using a face-detection model to address appropriate transformations of the identity.

---

### Official Review · Reviewer_rEcq · 2025-11-02

**Soundness:** 2
**Presentation:** 2
**Contribution:** 1
**Rating:** 2
**Confidence:** 3

**Summary:**

This paper introduces PiQPerfect, a diffusion-based method for same-identity facial replacement—replacing a person’s face in one image with another photo of the same person, while maintaining the background and overall realism.
The authors fine-tune SDXL using a distorted inner-face conditioning strategy that avoids trivial copying and introduce a large-scale synthetic dataset of ~500K identity-consistent image pairs generated with LivePortrait.
They report superior performance over state-of-the-art face reenactment and identity-preserving diffusion methods on both in-distribution and out-of-distribution benchmarks.

**Strengths:**

The focus on same-identity face replacement—as opposed to generic reenactment or swapping—is well motivated and practically relevant for photography and media applications.

**Weaknesses:**

1)  The proposed method primarily fine-tunes SDXL using spatial concatenation and a distorted-face conditioning input. These ideas are reminiscent of FaceAdapter, Instant-ID and IP-Adapter . While the task setup differs, the technical core offers only an incremental variation on prior conditioning strategies.

2) The PiQPerfect dataset is generated entirely via LivePortrait, raising concerns about domain shift and realism. There is no evaluation demonstrating generalization to real photographs.


3) Several baselines (FADM, FaceAdapter, HyperReenact) are designed for cross-identity reenactment and are not retrained for the same-identity setup, making comparisons less meaningful. Moreover, reported FID failures for some baselines compromise statistical validity.


4) The main metrics (FID, LPIPS, PSNR) do not reliably correlate with perceptual realism, and the user study (10 participants × 25 samples) lacks statistical power.

5) The ablation study focuses on architecture variants but does not isolate the effect of the distorted-face conditioning or quantify how each augmentation (relighting, masking) contributes.


6) The work directly intersects with face manipulation and identity-based generation, yet ethical safeguards are minimal. The “Potential Social Impact” section is cursory and limited to watermarking.


7) The main text spends disproportionate space on dataset construction and training details. The core ideas could be expressed more succinctly.


8) Claims of applicability to “group photo editing” are not convincingly demonstrated on real-world data.

**Questions:**

Generalization to Real Photos: The dataset and most evaluations are synthetic, generated using LivePortrait. Can the authors provide quantitative or at least visual results on real, non-synthetic photographs (e.g., celebrity photo datasets, mobile camera bursts, or natural photo pairs)?

Effect of the “Distorted Face” Conditioning: The main novelty is the use of a distorted version of the target face as a driving signal. Could the author comment on  (a) direct use of the unmodified target face, (b) random noise or masking instead of distortion, and (c) varying levels of distortion intensity?

Baseline Re-training: Were any baselines retrained or fine-tuned on your same-identity dataset (e.g., FaceAdapter, FADM)? If not, how do the author justify direct comparison to models trained for cross-identity reenactment?


Identity Preservation Metrics
The papers rely primarily on ArcFace cosine similarity (CSIM) and FID. Have you evaluated identity consistency with multiple face embedding models (e.g., AdaFace, ElasticFace, or MagFace) to verify robustness to embedding bias?

User Study Design
Could the authors provide more details about your user study (participant demographics, evaluation interface, instructions, and randomization)?


Ethical Data Collection and Consent
How did the authors ensure informed consent and proper licensing for the videos used to generate the PiQPerfect dataset? Were participants compensated, and will consent documentation be released with the dataset?

Watermarking and Misuse Prevention
The authors mention embedding digital watermarks as a mitigation step. Have you implemented or tested this? If so, how does the watermark survive downstream transformations (cropping, compression)?

Failure Modes Analysis
The appendix briefly mentions failures around eyes and hair. Can you categorize these errors (e.g., illumination mismatch, misalignment, missing details) and quantify their frequency?

Computational Efficiency and Latency
The paper states inference takes ~3 seconds at 576×576 on an A100 GPU. How does this scale to higher resolutions (e.g., 1024×1024) and to consumer hardware?

Public Dataset Release Plan
The paper stated the dataset “will be released publicly.” Could the authors clarify under what license, with what restrictions, and whether raw video data or only generated images will be released?

---

> ### Author Response · Authors · 2025-11-14
>
> We thank the reviewer for the feedback and for recognizing the applicability and practical relevance of our solution. Below we address the main criticisms.
>
> ---
>
> > About *“generalization to real photographs”*
>
> We acknowledge the importance of this concern and believe that this needs further clarification. Despite the entire training dataset being synthetically generated, **half of the “out-of-distribution” test-set is real-world data**. We followed a different process for the 5,000 PiQPerfect test-images in this setup, where two random frames were captured from two separate videos submitted by the same identity person. The results on these test-sets ensure that **the model generalizes well to real-world as well as synthetically generated data**. Additionally, the example shown in Figure 9 showcases another potential real-world application of our solution to composite photograph editing. The 4 photographs on the left half of the figure were taken in a “burst-mode” setup, while the one on the right is the result of applying our model to the faces of the input (top-left image), and using faces from the other pictures (highlighted in colored boxes) as driving images.
>
> ---
>
> > About *“ideas being reminiscent of FaceAdapter, Instant-ID and IP-Adapter”*
>
> We respectfully disagree with the statement and believe that our architecture differs from the ones mentioned by not having an “adapter” component.
>
> ---
>
> > About *“comparisons with cross-identity reenactment baselines being less meaningful”*
>
> While we agree that cross-reenactment and same-identity face replacement are distinct tasks, we disagree with this criticism. To our knowledge, there are **no publicly available baselines tailored for same-identity face replacement**. In the absence of direct baselines, we selected the most capable publicly available methods that operate on similar inputs and applied them in the same-identity setup. Importantly, all compared methods receive the same inputs, including full driving faces. Our evaluation setup does not disadvantage competing methods.
>
> ---
>
> > About *“unreliability of metrics and lack of statistical power of the user study”*
>
> We disagree with the statement and believe that our approach is clearly better than the other baselines. We have enough samples to conclude the significant performance gap between our solution and other benchmarks. We additionally report that each participant in the user study was instructed to annotate 25 images as “real” or “fake” for each model and for each setup (ID and OOD), amounting to a total of 325 votes per user, i.e. 3250 data points.
>
> ---
>
> > About *“the effects of data augmentations and distorted-face conditioning”*
>
> We perform comprehensive ablation studies to evaluate the effects of data augmentation techniques and showcase our observations highlighting the impact of using augmentations during training in Figure 7, the effect of relighting in Figure 8, and the model’s robustness to these augmentations in Figure 13 of the appendix. We do not provide quantitative metrics on these studies since in these cases the model tends to overfit and cheat by copy-pasting the driving image thereby boosting all metrics and misleading conclusions.
>
> ---
>
> > About *“evaluating other face embedding models and their bias”*
>
> We use ArcFace for CSIM and FID as is a known practice in most academia related face reenactment work. Since these embeddings aren’t used in training, the model cannot possibly overfit ArcFace.
>
> ---
>
> > About *“details regarding dataset release and consent documentation”*
>
> Data was collected under direct agreements with crowd workers, permitting both research and commercial use and ensuring compliance with privacy regulations, including GDPR-equivalent standards. All videos were manually reviewed to exclude issues such as individuals in the background. Personally identifiable information and metadata were removed to ensure participant privacy. All contributors were fairly and appropriately compensated according to regional standards and signed consent forms explicitly permitting research and commercial use of their video and audio data, including training AI models. A contact email will be provided on the dataset release page for participants to request data removal at any time. Upon acceptance, we plan to release the dataset under a research friendly license. Specifically, we plan to release (neutral + expressive) face images as well as driving videos, in case models other than LivePortrait are preferred.

---

### Note · Authors · 2025-11-14

**Comment:**

We thank the reviewers for their time and appreciate the feedback received. We will take into account all the questions raised and submit an improved version of our work to a future conference.

**Withdrawal Confirmation:**

I have read and agree with the venue's withdrawal policy on behalf of myself and my co-authors.